# Enhancing Withanolide Production in the *Withania* Species: Advances in In Vitro Culture and Synthetic Biology Approaches

**DOI:** 10.3390/plants13152171

**Published:** 2024-08-05

**Authors:** Zishan Ahmad, Irfan Bashir Ganie, Fatima Firdaus, Muthusamy Ramakrishnan, Anwar Shahzad, Yulong Ding

**Affiliations:** 1State Key Laboratory of Tree Genetics and Breeding, Co-Innovation Centre for Sustainable Forestry in Southern China, Bamboo Research Institute, Key Laboratory of National Forestry and Grassland Administration on Subtropical Forest Biodiversity Conservation, School of Life Sciences, Nanjing Forestry University, Nanjing 210037, China; ahmad.lycos@gmail.com (Z.A.); ramky@njfu.edu.cn (M.R.); 2Department of Environmental Engineering, College of Biology and the Environment, Nanjing Forestry University, Nanjing 210037, China; shareenk328@gmail.com; 3Department of Botany, Aligarh Muslim University, Aligarh 202002, India; irfanbashir301@gmail.com (I.B.G.); ashahzad.bt@amu.ac.in (A.S.); 4Chemistry Department, Lucknow University, Lucknow 226007, India; fatimafirdaus74@gmail.com

**Keywords:** CRISPR, cell culture, machine learning, synthetic biology, withanolide

## Abstract

Withanolides are naturally occurring steroidal lactones found in certain species of the *Withania* genus, especially *Withania somnifera* (commonly known as Ashwagandha). These compounds have gained considerable attention due to their wide range of therapeutic properties and potential applications in modern medicine. To meet the rapidly growing demand for withanolides, innovative approaches such as in vitro culture techniques and synthetic biology offer promising solutions. In recent years, synthetic biology has enabled the production of engineered withanolides using heterologous systems, such as yeast and bacteria. Additionally, in vitro methods like cell suspension culture and hairy root culture have been employed to enhance withanolide production. Nevertheless, one of the primary obstacles to increasing the production of withanolides using these techniques has been the intricacy of the biosynthetic pathways for withanolides. The present article examines new developments in withanolide production through in vitro culture. A comprehensive summary of viable traditional methods for producing withanolide is also provided. The development of withanolide production in heterologous systems is examined and emphasized. The use of machine learning as a potent tool to model and improve the bioprocesses involved in the generation of withanolide is then discussed. In addition, the control and modification of the withanolide biosynthesis pathway by metabolic engineering mediated by CRISPR are discussed.

## 1. Introduction

In ancient literature, the usage of plants as a source of medicine has been documented [1]. Medicinal plants and their secondary metabolites have garnered renewed interest for several compelling reasons, rooted in their accessibility, cost-effectiveness, efficacy, and safety profile [2]. These factors make them an attractive option for developing new therapeutics and healthcare solutions. In 2017–2018, the global market for enhanced-value extracts of herbal products and medicinal plants reached $456.12 million, a 12.23% increase from the previous year [3]. The herbal medicine market is projected to increase at a 5.88% compound annual growth rate (CAGR) and reach USD 129,689.3 million by 2023. Currently, plants and plant extracts provide the majority of healthcare for almost 80% of the world’s population. The World Health Organization (WHO) predicted that by 2050, the global herbal market would be worth $5 trillion [3]. 

There is a huge potential for developing mass cultivation systems for plant-based medicines in the future due to the global shift towards these medicines, their extraordinary demands, and the accompanying trades. *Withania* is a genus that is part of the Solanaceae family. It has 61 species that may be found worldwide [4], among which *Withania somnifera* (L.) is well studied. Due to its extensive medical qualities, which include anti-cancer, cardioprotective, anti-inflammatory, immunomodulatory, anti-coagulant, anti-diabetic, anti-oxidant, and neuroprotective effects, *W. somnifera* holds a unique position in the genus *Withania* [5]. Because of their many pharmacological characteristics, steroidal lactones, specifically withanolide A, withanolide D, withanolide E-M, withaferin A, and withanone, have drawn the interest of researchers [6,7]. *Withania* species, particularly *W. somnifera*, are of significant importance due to their rich content of withanolides, which have broad-spectrum therapeutic properties. The increasing market demand for natural remedies and the proven medicinal benefits of withanolides justify the need for advanced cultivation and production systems for *Withania* species to meet global needs.

Next-generation sequencing (NGS) has made major advances recently, opening the door to the characterization and identification of biosynthesis pathways. As a matter of fact, multi-omics investigations (such as transcriptomics, epigenomics, proteomics, and metabolomics) have led to a clearer understanding of the metabolic pathways for a wide range of secondary metabolites [8]. Furthermore, the particular roles of regulatory elements and genes in the production of secondary metabolites can be precisely understood through the use of in silico methods for gene and gene cluster predictions involved in the biosynthesis of secondary metabolites, especially through sophisticated computational techniques like machine learning (ML), followed by detailed molecular characterization using genome and epigenome editing tools [9]. Utilizing these data, genetic engineering is capable of producing synthetic gene clusters that can be inserted into in vitro plant cultures (like cell or hairy root cultures) or microorganism cultures (like yeast and bacteria) to promote the production of particular secondary metabolites. Furthermore, hairy root and/or cell suspension cultures can benefit from the metabolic optimization achieved using CRISPR-mediated targeted enzyme engineering techniques, increasing secondary metabolite production [10]. The identification of gene regulatory networks connected to commercially important withanolides has been made possible by these developments. Thus, by combining genome editing techniques with in vitro growth techniques, the increasing demand for withanolide can be satisfied.

The proposed study includes an integrated assessment of promising conventional procedures such as the use of elicitors and precursors, optimization of media composition and condition, and new developments in in vitro culture-mediated withanolide production. The development of these metabolites’ synthesis in heterologous systems is then examined and emphasized. Subsequently, the use of machine learning (ML) to enhance comprehension of the withanolide biosynthesis pathway is explored, along with the modeling and optimization of bioprocesses (such as cell suspension culture, hairy root culture, and bioreactors) using ML-mediated system biology integration. Lastly, a review and discussion of the regulation and modification of withanolide production pathways by CRISPR-mediated metabolic engineering are provided. 

## 2. Withanolide Production through In Vitro Culture

In vitro culture offers significant advantages for plant propagation, conservation, and the production of secondary metabolites. Different techniques are being employed to produce withanolides from the genus *Withania* (Figure 1). The production of secondary metabolites through in vitro culture offers several significant benefits over conventional methods. By providing a controlled environment, in vitro cultures enable precise regulation of factors like light, temperature, nutrient availability, and hormonal balance, which can optimize metabolite production [11]. This method allows for year-round cultivation, ensuring a continuous supply of secondary metabolites regardless of seasonal and climatic variations that affect traditional farming. Moreover, in vitro cultures reduce the risk of disease and pest contamination by maintaining sterile conditions, thus ensuring the production of high-quality, uncontaminated compounds. Scalability is another advantage, as in vitro techniques can be easily scaled up using bioreactors, facilitating large-scale production [12]. This section is focused on the production of withanolides through different strategies. 

### 2.1. Cell Suspension Culture

Cell suspension culture plays a crucial role in plant biotechnology, providing a versatile tool for producing secondary metabolites, conducting fundamental research, conserving rare species, and exploring genetic engineering possibilities [13]. It offers an advantage over traditionally based means of secondary metabolite production owing to its potential to produce secondary metabolites on a large scale and harvest specific metabolites within a limited period of time [14]. To generate a cell suspension culture, the callus formed on a solid medium is typically transferred to a liquid medium. Unlike solid media cultures, where cells are anchored to a surface or embedded in a gel, cell suspension cultures allow plant cells to proliferate freely in a liquid environment. Cell suspension culture, while highly beneficial for various biotechnological applications, carries several risks that can impact its efficiency and reliability. For example, this methodology is linked to slower growth rates and lower metabolite outputs when compared to other secondary metabolite synthesis methods, such as microbial cultures [12]. Nonetheless, there are strategies to speed up the synthesis of secondary metabolites, including altering and optimizing the media compositions, precursor feeding, using elicitors, and modulating ambient factors. For example, a comparative analysis of different media and growth regulators (MS, B5, NN, N6, and auxins, and a combination of auxin and cytokinin, respectively) and different carbon sources (sucrose, glucose, fructose, and maltose) for obtaining cell suspension cultures of *W. somnifera* was carried out [15]. They optimized conditions to maximize biomass accumulation and withanolide A production, identifying the ideal parameters as full-strength MS medium, 3% sucrose, a four-week culture period, and an initial pH of 5.8 [15]. Similarly, the highest biomass and withanolide production were observed at optimized additive concentrations of L-glutamine in combination with picloram and Kn in *W. somnifera* [16]. Plant growth regulators (PGRs) play a crucial role in the development and optimization of cell suspension cultures for the production of withanolides. For example, maximum withanolides were obtained when MS media was supplemented with the optimized concentration of IAA and BA when supplemented in combination [17]. However, the nutrient composition of the culture medium must be optimized in conjunction with PGRs. The availability of essential nutrients can significantly influence the effectiveness of PGRs on withanolide production, so it is crucial to find the optimal level of media components. To overcome the current restrictions and difficulties in using PGRs for withanolide production and pave the way for more effective, scalable, and sustainable production methods, research is needed on the synergistic effects of PGRs with different nutrient compositions and supplements in the culture medium.

Precursor feeding is a biotechnological strategy used to enhance the production of specific secondary metabolites in cell cultures by supplying precursor molecules directly to the culture medium [18]. Precursor feeding in cell suspension cultures for enhanced production of secondary metabolites like withanolides is often limited due to several factors. Optimizing the concentrations and timing of precursor addition requires a detailed understanding of the metabolic pathways involved. However, these pathways are not well-characterized for the genus *Withania*. In addition, identifying suitable precursors that can effectively stimulate the biosynthetic pathways without causing cytotoxic effects or metabolic imbalances can be challenging. In a study, the optimization of major and minor withanolides in *W. somnifera* was carried out using elicitors and precursor feeding strategies [19]. The highest withanolide yields were achieved with chitosan and squalene, along with picloram, Kn, L-glutamine, and sucrose, in bioreactors. This protocol resulted in significantly higher withanolide concentrations compared to controls, both in shake-flask and bioreactor cultures. Precursor feeding presents difficulties due to metabolic complexity, expense, optimization challenges, and potential toxicity. Nevertheless, these obstacles may be overcome with further study and technical advancements. Subsequent developments in metabolic engineering, biotechnological instruments, and industrial use are anticipated to broaden the range and significance of precursor feeding. These advancements will play a crucial role in augmenting the yield of important chemicals generated from plants.

Elicitors are another potentially useful strategy in biotechnological applications when the goal is to increase the variety and production of secondary metabolites, including withanolides. This approach leverages the plant’s natural defense mechanisms to enhance the production of bioactive compounds in cell cultures. The secondary metabolites in plants are fundamentally triggered by external environmental agents during their growth under in vitro conditions. The external agents are called elicitors and are classified as biotic and abiotic elicitors. To increase the production of secondary metabolites in various plant species, a variety of elicitors are being used, including chitosan, yeast extract, jasmonic acid (JA), salicylic acid (SA), abscisic acid (ABA), methyl jasmonate, inorganic salts, heavy metals, physical agents like UV radiation, and microbes like bacteria and fungi [20]. In a recent study, the effects of SA and cellulase from *Aspergillus niger* were studied in *W. coagulans* cell suspension cultures, wherein SA was found to be more effective and promote higher withanolide accumulation than cellulase [21]. The results revealed that both elicitors at all durations of treatment promoted total phenol and flavonoid content, withaferin A, and withanolide A accumulation in leaf- and stem-derived cultures. However, higher levels of accumulation for withaferin A and withanolide A were observed in the cell suspension cultures derived from the leaf explant in comparison to the stem-derived cultures [21]. Similar research was conducted on the impact of macroelements and a nitrogen source on the accumulation of biomass and the generation of withanolide A from *W. somnifera* cell suspension cultures [15]. It was observed that cell suspension culture produced a larger biomass when the medium was augmented with NH_4_NO_3_, and the highest withanolide content was produced when the medium was augmented with KNO_3_. These results indicate that nitrate and ammonium ions have different effects on the metabolic pathway in plants [15]. The use of biotic elicitors was also found to be beneficial in promoting the accumulation of withanolides. For example, a significant enhancement of withanolides was recorded in the cell suspension cultures of *W. somnifera* upon the addition of cell homogenates of different fungal species—*A. alternata*, *F. solani*, *V. dahliae*, and *P. indica* [22]. Despite the advantages they offer, these elicitors have drawbacks regarding large-scale withanolide production. Crucially, the biosynthetic pathways that elicitors activate are significantly influenced by the interplay between the elicitors and the availability of metabolic precursors. In addition, the interplay between elicitors and PGRs is crucial and may work synergistically or antagonistically, affecting overall metabolite production and cell growth [23].

Environmental factors play a crucial role in the production of secondary metabolites in plants, including withanolides. Light, temperature, water, soil composition, salinity, and air quality are some abiotic stresses that affect the production of secondary metabolites. Understanding these factors and their interactions can help optimize conditions for the enhanced production of valuable secondary metabolites, both in natural settings and in controlled environments like in vitro cultures. For example, short-term UV-B exposure notably enhanced the content of withaferin A and withanolide A, along with the overexpression of related biosynthesis genes in *W. coagulans* [24]. The results suggest that UV-B can effectively elicit secondary metabolism, improving the production of valuable pharmaceutical compounds in *W. coagulans*. In *Physalis peruviana* shoot cultures, the effects of mild heat stress (45 °C for 2 and 5 h) and UV-B radiation (313 nm for 15 min and 3 h) were also investigated [25]. The results showed a considerable increase in withanolide accumulation and squalene synthase gene expression. In a different investigation, red laser light treatment significantly increased the amount of withanolide in *W. somnifera* seedlings [26]. Salinity stress can induce the production of secondary metabolites in plants by triggering physiological and biochemical responses. This stress stimulates pathways involved in secondary metabolite biosynthesis as part of the plant’s defense mechanism. 

For example, a range of NaCl concentrations significantly increased withaferin A, withanolide A, and withanone levels in shoots and roots compared to the control in *W. somnifera* [27]. NaCl stress significantly enhanced withaferin A (3.79 mg/g), withanolide A (0.51 mg/g), and withanone (0.022 mg/g) at 50 mM NaCl groups in the shoot. Similarly, in the root, a significant increase in WFA (0.19 mg/g) and WN (0.0016 mg/g) were observed at 10 mM, WA (0.059 mg/g) at 30 mM, and withanolide B (WB) (0.013 mg/g) at 50 mM NaCl groups compared to the control [27]. By upregulating important biosynthetic genes under NaCl stress, melatonin promoted the production of withanolides, which resulted in an increase in the accumulation of withanolides A and withaferin A up to 1.6 µg/g and 14 µg/g, respectively, in *W. coagulans* [28]. Furthermore, the *W. somnifera* CAS gene is essential for controlling the production of phytosterols and withanolides, and it exhibits differential regulation in response to diverse abiotic stressors [29]. Understanding this regulation is vital for optimizing the production of these valuable compounds in Ashwagandha and enhancing the plant’s medicinal properties and stress resilience. One of the most significant stresses that may be brought on by osmotic or drought stressors is water scarcity or availability. It is simple to induce water stress by simply adding polyethylene glycol to the medium as a high-molecular-weight solute [30]. Numerous medicinal plants have already shown the beneficial effects of water stress on the formation of secondary metabolites in vitro; however, the study is limited to the genus *Withania*. A study on *W. somnifera* under drought stress revealed decreased photosynthetic parameters and increased expression of genes related to osmoregulation, detoxification, and secondary metabolite production [31]. The findings suggest potential pathways for developing drought-tolerant genotypes with enhanced secondary metabolite production. Seasonal fluctuations have a major impact on the production of withanolide and on plant development and productivity. Wintertime is characterized by higher expression of transcription factors and important pathway genes related to the growth of plants and the synthesis of secondary metabolites [32]. The observations highlight how environmental influences affect the production of secondary metabolites and how transcription factors regulate *W. somnifera*.

The aforementioned investigations seek to improve the synthesis of withanolides, a valuable secondary metabolite, by comprehending and adjusting stress responses and environmental conditions. Researchers can improve conditions to boost the output of withanolides by understanding how these abiotic and biotic factors impact the biosynthesis of these chemicals. This information is essential for regulated in vitro settings as well as natural cultures, which will ultimately increase the therapeutic value and stress tolerance of plants of the genus *Withania*. However, due to natural variability, results produced in controlled settings might not transfer directly to field contexts, and species-specific responses might restrict generalizability. The intricate relationships between many environmental elements are frequently overlooked, and the long-term effects of stress applications still need to be understood. Advanced methods and resources are needed for comprehensive gene expression analysis, yet it still needs to be improved to balance growth with metabolite synthesis. Table 1 represents the use of different elicitors for the enhancement of the withanolides.

### 2.2. Hairy Root Culture 

One useful method for producing secondary metabolites that are first biosynthesized in the roots or even in the aerial organs of mature plants is the induction of hairy root culture by *Agrobacterium rhizogenes* [38]. It has the potential to significantly boost the output of secondary metabolites by combining the use of elicitors and other methods with pathway gene manipulation. T-DNA from a root-inducing plasmid (Ri plasmid) of *Rhizobium rhizogenes* is transferred into plant cells to induce hairy roots, which lead to the prolific growth of neoplastic roots. Due to their quick growth rate and genetic stability, hairy root cultures have an array of benefits, one of which is their capacity to generate large amounts of secondary metabolites [38]. This technique eliminates the requirement for entire plant cultivation by offering a consistent and sustainable source of important chemicals. Furthermore, hairy root cultures are simple to maintain and work with in vitro, making it possible to precisely manage the genetic alterations and growth circumstances. However, the initial complexity of producing the culture, which necessitates the effective transformation of plant tissues with *A. rhizogenes*, is one drawback [39]. Depending on the species of plant and kind of tissue, the procedure might be species-specific and have different success rates. Despite these difficulties, hairy root cultures offer a strong foundation for researching plant biology and generating secondary metabolites once they are established. 

Several attempts have been made to enhance the production of withanolide in the plants of *Withania* (Table 1). For example, *A. rhizogenes* causes cotyledons and leaf explants to induce the growth of hairy roots in *W. somnifera* [40]. Southern analysis and PCR were used to validate the transgenic status of the *rol*B and *npt*II genes. All hairy root clones grown in MS-based liquid medium and maximum withanolide were recorded in transformed roots compared to non-transformed roots [40]. In another study on *W. somnifera* hairy roots, based on the accumulation of bioactive compounds and development rate, seven superior hairy root lines were chosen. The synthesis of withanolides and phenolic acids was markedly increased by elucidation with salicylic acid [41]. The elicitation of withanolide production in 150 hairy root lines of *W. somnifera* was investigated using targeted elicitation strategies [42]. The W9 line was the focus of this analysis due to its high growth rate and withanolide content. MeJ and *P. indica* cell homogenate were examined at different doses and times, and the greatest amounts of withanolide were obtained at 15 μM MeJ for 4 h [42]. The focus on a single hairy root line and possible response heterogeneity between lines or circumstances are the study’s weaknesses, though. Successful transformation was achieved in *W. somnifera* using *A. rhizogenes*, wherein half-MS liquid medium was optimal for hairy root growth [43]. Secondary metabolite production, particularly withaferin A, was enhanced through various methods, with *Aspergillus* homogenate showing positive results. However, varying transformation efficiencies and lower withaferin A content in in vitro roots compared to field roots are the limitations of the study. An investigation was carried out on the impact of macroelements and nitrogen sources on biomass and the formation of withanolide A in the hairy roots of *W. somnifera* [43]. KNO_3_ yielded the largest generation of withanolide A, although KH_2_PO_4_ produced the most biomass. The ideal ratios of NH_4_^+^ to NO_3_ for nitrogen sources were also determined in order to maximize biomass and produce withanolide. The results support the large-scale cultivation of *Withania* hairy roots for the manufacture of withanolide A [43]; nevertheless, more research into other growth factors and the long-term stability of the ideal circumstances is needed. 

Another technique that did not include *A. rhizogenes* produced withanolides in *Withania* plants’ adventitious roots grown from callus by employing both biotic and abiotic elicitors [35]. In MS media containing IBA and NAA, calli from leaf explants generated more roots, and secondary roots multiplied. As elicitors, chitosan (biotic) and aluminum chloride (abiotic) were employed; withanolide synthesis was greatly increased by chitosan [35]. While considerable biomass accumulation and a 21-fold increase in withanolide A content compared to natural leaves were seen, the impact of PGRS and sucrose on adventitious root development from leaf explants of *W. somnifera* to boost withanolide A production was achieved [36]. In addition, the impact of a carbon source on high biomass and withanolide A was examined, and the results showed that 2% sucrose concentration was optimal for both biomass and secondary metabolite formation in adventitious roots of *W. somnifera* [44]. Using MeJ and SA as elicitors, improved adventitious root cultures from *W. somnifera* were achieved for increased withanolide synthesis. Withanolide A, B, withaferin A, and withanone production were all markedly increased by SA, along with the generation of major and minor withanolides [45]. 

When it comes to producing secondary metabolites, hairy root cultures are preferable to adventitious root cultures in a number of ways [46]. Some benefits include, but are not limited to, rapid growth, genetic stability, and the capacity to produce large amounts of secondary metabolites that resemble those in the parent plant. Furthermore, the generation of secondary metabolites mediated by hairy root culture can be further enhanced by the use of both biotic and abiotic elicitors [46]. However, there are multiple challenges to producing secondary metabolites on a large scale using hairy root culture, including maintaining uniform growing conditions and protecting against contamination. By offering a regulated environment for ideal root development and permitting precise management of nutrients, oxygen, and other factors, bioreactors help to address these problems. Additionally, they help maintain sterility, lowering the possibility of contamination and enabling the generation of secondary metabolites in a scalable and repeatable manner. Different kinds of bioreactors have been suggested, such as gas phase, liquid phase, and hybrid systems, for the synthesis of secondary metabolites [47]. Various medicinal plants have been employed in bioreactors for the production of secondary metabolites on a larger scale, such as *Stevia rebaudiana* [48], *Catharanthus roseus*, and *Santalum album* [49]. The scale-up process is still impacted by a number of variables, though, and they should be considered and adjusted. The variables include medium composition, growth kinetics, culture time, inoculum density, and gaseous composition [50]. However, the use of such bioreactors for the production of withanolide is very limited. In an investigation that used precursor feeding and elicitation methods in shake-flask and liquid phase bioreactor settings to maximize the production of major and minor withanolides in *W. somnifera* cell suspension cultures, these methods were examined [19]. Future research pertaining to the synthesis of withanolides mediated by hairy root culture must prioritize the development of innovative bioreactors and culture containers. This involves understanding the dynamic interplay of elicitors, nutrients, and growth conditions to maximize withanolide yields without compromising cell viability or product quality. Exploring genetic and metabolic engineering approaches could enhance the metabolic pathways responsible for withanolide biosynthesis, potentially increasing yields and diversifying product profiles. Moreover, developing integrated bioprocess strategies that consider the entire production chain, from culture inoculation to downstream processing and product purification, will be crucial for achieving economically viable and sustainable production of withanolides in bioreactor systems.

## 3. Withanolide Production via Synthetic Biology: Innovations and Challenges

Through the use of engineering concepts, the multidisciplinary discipline of synthetic biology produces new biological systems or redesigns existing ones for practical uses. It entails the conceptual engineering and design of biological systems, circuits, and organisms to accomplish certain goals. Synthetic biology has made recent advancements that have revolutionized several fields. It offers significant economic benefits for secondary metabolite production by enabling the development of highly efficient and scalable bioproduction platforms. Traditional methods of extracting secondary metabolites from natural sources often involve extensive cultivation, long growth periods, and complex extraction processes, which can be costly and time-consuming. Synthetic biology, however, leverages genetic engineering and metabolic pathway optimization to construct microbial- or plant-based cell factories that can produce these valuable compounds more efficiently [51]. This approach not only lowers production costs but also ensures a consistent and sustainable supply of metabolites. Additionally, synthetic biology enables the production of novel or rare compounds that are difficult to obtain from natural sources, opening up new market opportunities [52]. Our understanding of the processes involved in withanolide production has increased due to recent developments in the molecular biology of *Withania* plants. Advances in this subject offer the chance to set all of the biosynthetic pathway’s genes together in one artificial gene cluster [53]. Plant metabolic diversity reflects adaptation to environmental niches, with many compounds having crucial ecological roles. Gene clusters involved in plant biosynthesis are critical in generating a variety of compounds required for development and defense against stress. The development of multi-omics technology has improved the mining of these clusters, providing fresh perspectives on the variety of secondary metabolites [54]. These biosynthetic gene clusters show various structural arrangements, and their formation and evolution are influenced by broader metabolic, population, and epigenetic factors [53].

Several essential elements are needed for synthetic biology to produce secondary metabolites rapidly. First, since these gene clusters include the genes required for metabolite production, it is essential to identify and characterize the biosynthetic gene clusters found in the genomes of plants and microbes [55]. CRISPR-Cas9 and other cutting-edge genetic engineering techniques are crucial for exact alterations and pathway improvement [56]. To increase output, efficient expression systems and host organisms, such as plant cell cultures or modified microorganisms, are required. Furthermore, to maximize production and guarantee cost-effectiveness, regulatory element fine-tuning, fermentation process scaling, and metabolic pathway optimization are essential [56]. Ultimately, the process can be further expedited and improved by employing biosensors and computer modeling to provide continuous monitoring and adaptive control of manufacturing parameters.

The common method for fast functional characterization of genes involved in secondary metabolite biosynthesis pathways is the use of transient expression and heterologous systems. Multiple plants, for example, *Nicotiana benthamiana*, *A. thaliana*, *Solanum lycopersicum*, *N. tabacum*, *E. coli*, and yeast, can be used, with the choice depending on the ease of transformation and the relevance to the metabolite of interest. In this section, we will discuss the recent advancements in withanolide production through plant cell heterologous systems. Table 2 depicts the use of synthetic biology for withanolide production.

### 3.1. Withanolide Production through Heterologous and Cell-Free Systems 

Recent advancements in synthetic biology have spurred interest in using plant cell heterologous systems for producing high-value plant-specific secondary metabolites [18]. Despite challenges like post-translational modifications and substrate availability, these systems offer a safe, cost-effective, and eco-friendly approach. However, gene transformations and CRISPR-Cas9 editing provide exciting new avenues for the expansion of biosynthetic pathways and the synthesis of metabolites in the future [18]. To enhance the production of secondary metabolites, including withanolides, in plant cells in vitro, *Agrobacterium*-mediated gene transfer accompanied by modification of secondary metabolite biosynthesis pathways has been extensively utilized. For example, *WRKY1* from *W. somnifera* sheds light on its pivotal role in the biosynthesis of triterpenoid withanolides via the phytosterol pathway [61]. It has been found that *WsWRKY1* controls the expression of important genes in the phytosterol pathway, including squalene synthase and squalene epoxidase. *WsWRKY1* is crucial for the production of phytosterols and withanolides, as evidenced by the decreased plant growth and phytosterol and withanolide levels seen in *W. somnifera* after its silencing. On the other hand, *WsWRKY1* overexpression increased tomato and tobacco phytosterol levels in addition to increasing *W. somnifera*’s synthesis of triterpenoid compounds. Furthermore, it was shown that overexpressing *WsWRKY1* in tobacco led to enhanced tolerance to bacterial, fungal, and insect challenges, indicating its dual role in metabolic engineering for boosting triterpenoid biosynthesis and strengthening plant defensive systems [61]. These results highlight the potential of *WsWRKY1* as a useful tool for biotechnological applications intended to improve the synthesis of pharmaceutical compounds and the ability of plants to withstand biotic stressors. Similarly, overexpression of *WsMYC2* increased the mRNA transcript levels of key pathway genes (*WsCAS*, *WsCYP85A*, *WsCYP90B*, and *WsCYP710A*), correlating with enhanced production of withanolides and stigmasterol [62]. Conversely, silencing *WsMYC2* reduced the levels of withanolides and stigmasterol, confirming its pivotal role in their biosynthesis. Additionally, the study identified cis-regulatory elements in the upstream promoter of *WsMYC2* associated with phytohormone responsiveness, specifically MeJA, which induced *WsMYC2* expression and subsequent metabolite accumulation. These findings highlight *WsMYC2* as a key regulator in the complex pathway leading to withanolide biosynthesis in *W. somnifera* [62].

In a different investigation, the cytochrome P450 (*CYP450*) genes *WsCYP749B1*, *WsCYP76*, and *WsCYP71B10* were shown to exhibit enhanced expression upon treatment with MeJ, suggesting their role in secondary metabolism [63]. Withanolide accumulation was significantly impacted by both overexpression and the maximum expression found in leaves, as determined by VIGS characterization. *WsCYP76* overexpression improved withanolide A production, but *WsCYP749B1* overexpression significantly raised withanolide A and B levels. *WsCYP71B10* silencing decreased the amount of withanolide B. Furthermore, in transgenic tobacco and *W. somnifera*, overexpression of these genes enhanced tolerance to *Pseudomonas syringae*, highlighting their dual function in withanolide production and biotic stress response. These results demonstrate how *CYP450* manipulation may be used to improve the synthesis of therapeutic compounds and plant resistance to infections in Ashwagandha and related species [63]. To completely comprehend CYP450’s participation in withanolide production and stress tolerance mechanisms, however, more validation of their actions in natural environmental settings is necessary due to the intricacy of metabolic pathways involving many enzymes. Conversely, the impact of the mevalonate (MVA) and 2-C-methyl-d-erythritol-4-phosphate (MEP) pathways on the generation of sterol and withanolide was investigated in relation to the overexpression of *WsHMGR2* and *WsDXR2* in tobacco [57]. While WsDXR2 transgenic lines had greater cholesterol reduction, *WsHMGR2* overexpression resulted in higher amounts of cycloartenol, sitosterol, stigmasterol, and campesterol. These results were further supported by transient suppression and pathway inhibition tests, which indicated a preferred carbon translocation via the MVA pathway for the production of withanolides, especially through the campesterol/stigmasterol route. Withanolide content was not increased by cholesterol feeding, highlighting different metabolic pathways and their functions in the formation of secondary metabolites in plants [57]. It was also suggested that SMT1, a sterol-C24-methyltransferase reliant on S-adenosyl L-methionine, has a role in the production of withanolides in *W. somnifera* [64]. Although SMT1 is known to have a role in phytosterol biosynthesis in other plants, its precise function in *W. somnifera* withanolide production has not yet been investigated. They found that SMT1 in *W. somnifera* was suppressed by RNA interference (RNAi), which resulted in lower levels of campesterol, sitosterol, and stigmasterol and higher cholesterol. On the other hand, *N. tabacum* overexpressing SMT1 increased phytosterol levels without changing cholesterol. These results demonstrate the critical function of SMT1 in directing metabolites for withanolide biosynthesis, namely through the campesterol and stigmasterol pathways, and provide insight on the regulatory significance of SMT1 in the synthesis of therapeutic compounds from Ashwagandha [64]. *WsCYP71B35*, a cytochrome P450 enzyme from *W. somnifera*, was identified by its expression in tissues that accumulate withanolide and its induction by MeJ [58]. *WsCYP71B35* catalyzes the synthesis of withaferin A, withanolide A, withanolide B, and withanoside IV, but not phytosterols or 24-methylene cholesterol, according to biochemical tests conducted in yeast microsomes. Withanolide levels were changed in *W. somnifera* leaves and confirmed by VIGS and *WsCYP71B35* overexpression, indicating its involvement in their production. Furthermore, in *W. somnifera* and transgenic tobacco, *WsCYP71B35* overexpression increased resistance to *P. syringae* infection, underscoring its dual function in the generation of secondary metabolites and plant defense [58].

The studies presented underscore the potential of harnessing plant cell heterologous systems for enhancing the production of withanolides and other secondary metabolites. However, these advancements come with limitations. Utilizing tobacco as a heterologous system, while effective for proof-of-concept studies, presents challenges such as the potential for regulatory restrictions due to its association with tobacco products and differences in post-translational modifications compared to target plants [65]. Furthermore, achieving optimal yields of complex secondary metabolites in heterologous systems remains a technical challenge, influenced by factors like substrate availability, protein folding efficiency, and metabolic pathway complexity. Overall, while the findings highlight promising avenues for improving withanolide production through genetic manipulation in plant cell heterologous systems, ongoing research is needed to address technical challenges and validate these approaches in diverse environmental contexts. Balancing the advantages of increased production efficiency and regulatory compliance with the drawbacks of technical complexity and ethical considerations will be crucial for realizing the full potential of these biotechnological strategies in medicinal plant research and industrial applications.

A cell-free enzymatic platform for withanolide production refers to a biotechnological approach that utilizes isolated enzymes, rather than whole cells, to catalyze the biosynthesis of withanolides. This method involves extracting and purifying the necessary enzymes from their natural sources or producing them through recombinant DNA technology. Then, by means of certain biochemical processes, these enzymes are employed in vitro (apart from live cells) to transform precursor molecules into withanolides. Enhanced control, simplified pathway engineering, the elimination of cellular constraints, and scalability are some of the advantages [66]. However, integration and balancing of multiple enzymatic reactions and the production of precursor molecules and cofactors at sufficient levels are the main challenges that are associated with a cell-free system [66]. While cell-free systems offer enhanced control, they also lack the self-regulatory mechanisms of living cells, which can sometimes compensate for fluctuations in reaction conditions or substrate availability [67]. Studies on the use of cell-free platforms in *Withania*, however, are lacking. Nonetheless, this technique is effectively used to produce cannabinoids in *Cannabis* [68], where they developed a fed-batch enzymatic reactor system for high-titer production of CBGVA and CBGA from glucose, optimizing the use of minimally expensive molecules like CoA and ATP. They successfully demonstrated the conversion of prenylated substrates to CBDA and CBDVA in a single enzymatic step [68]. Withanolides are complex secondary metabolites, and their biosynthetic pathways involve multiple steps with numerous enzymes, making it challenging to replicate these processes outside of a living cell. Furthermore, technological advancements in enzyme engineering, pathway optimization, and reaction condition control may still be needed to make cell-free production of withanolides feasible and commercially viable.

### 3.2. Withanolide Production through a Microbial Heterologous System 

For extended periods of time, sustained enzymatic activity requires enzyme stability outside of its original cellular environment. This issue can be resolved in microbial heterologous systems. The high expenses of cofactor replenishment and enzyme purification in cell-free systems frequently make them less cost-effective. However, in microbial systems, genetic engineering might possibly reduce these costs by optimizing cofactor recycling pathways and enzyme synthesis. Furthermore, microbial heterologous systems can use metabolic engineering and genetic modification to precisely adjust metabolic pathways [69]. Therefore, in order to improve the scalability, efficiency, and economic viability of producing complex molecules such as withanolides, microbial heterologous systems provide complementary options. Microbial heterologous systems offer a promising platform for sustainable and efficient production of withanolides with diverse applications in pharmaceuticals, nutraceuticals, and other industries. In addition, it increases our understanding of the biosynthetic pathways involved in these complex secondary metabolites. By transferring and expressing key biosynthetic genes from *Withania* species or related plants into microbial hosts such as *E. coli* or yeast, researchers can elucidate the precise enzymatic reactions and regulatory mechanisms underlying withanolide biosynthesis. For example, cloning and characterization of a *W. somnifera* gene encoding squalene synthase (*WsSQS*) were conducted [59]. A 411 amino acid polypeptide was encoded by a 1236 bp ORF on the 1765 bp long, full-length cDNA that was extracted. Under ideal circumstances, recombinant *WsSQS* was produced in *E. coli*, with leaves exhibiting the greatest level of expression compared to other studied tissues. Promoter analysis revealed the presence of many cis-acting elements. SA, MeJ, and 2,4-D were shown to upregulate the biosynthesis of withanolides in accordance with the promoter’s expected regulatory elements [59]. The expression analysis of *WsSQS* across different tissues and the identification of regulatory elements in its promoter region provide insights into the spatial and temporal regulation of withanolide biosynthesis. This can lead to the discovery of new regulatory pathways and potential targets for metabolic engineering. While promoter analysis and expression studies of *WsSQS* across different tissues reveal regulatory insights, the actual manipulation of these regulatory elements for enhanced withanolide production remains underexplored. In another study, by interfering with the ERG4 and ERG5 enzymes in the yeast’s ergosterol pathway and substituting ERG5 with the 7-dehydrocholesterol reductase (DHCR7) enzyme, scientists were able to biosynthesize 24-methylene-cholesterol, a vital precursor for the production of physalin and withanolide [60]. They evaluated three DHCR7 versions from various organisms. In flask-shake culture, the *Xenopus laevis* DHCR7 (*XlDHCR7*) showed the best efficiency in generating methylene-cholesterol. This modified yeast strain provides a viable platform for more investigation into the downstream enzymes in the physalin and withanolide biosynthesis pathways, enabling more economical and productive synthesis of these important substances [60]. Although the use of *X. laevis* DHCR7 in yeast for methylene-cholesterol production shows promise, further optimization and characterization of downstream enzymes in the physalin and withanolide biosynthesis pathways are needed to maximize yields and ensure pathway robustness.

Since the formation of terpenoid and its precursors limits the entire biosynthesis of withanolides, further research should focus on engineering microbial hosts with optimized terpenoid biosynthesis pathways tailored for withanolide production. Exploring diverse microbial hosts and their genetic manipulation for enhanced precursor supply, such as mevalonate pathway optimization or the introduction of feedback-resistant enzymes, could significantly boost withanolide yields. High-yield withanolide manufacturing using microbial heterologous systems is still in its infancy. Additionally, investigating metabolic engineering strategies to balance flux towards withanolide biosynthesis while minimizing byproduct formation will be crucial. Thus, it is still possible to produce uncommon and/or small withanolides using genetically engineered microbes, which may then be further investigated and improved in the future. 

#### Biosynthesis Gene Clusters for Withanolides

Biosynthesis gene clusters are pivotal for the coordinated production of withanolides, housing the necessary genes for the entire biosynthetic pathway. These clusters typically encompass genes encoding enzymes responsible for precursor synthesis, structural modification, and the final assembly of withanolide molecules. Advances in genomic and transcriptomic studies have unveiled putative withanolide biosynthesis gene clusters in *W. somnifera*. These clusters generally contain genes for key enzymes such as squalene synthase (SQS), which initiates the pathway by converting farnesyl pyrophosphate (FPP) to squalene; cytochrome P450 monooxygenases, involved in the oxidation and structural diversification of withanolide intermediates; and glycosyltransferases, which catalyze the addition of sugar moieties to withanolide molecules, enhancing their solubility and bioactivity [59]. By transferring these entire gene clusters into microbial hosts like *E. coli* or yeast, researchers can reconstitute the complete biosynthetic pathway, facilitating the production of complex withanolides. This approach ensures coordinated gene expression, as the gene clusters promote the expression of all necessary enzymes together, thereby improving pathway efficiency. The presence of natural regulatory elements within these clusters can help optimize enzyme expression levels and activities, further enhancing yield. Additionally, clustering genes in close proximity can promote metabolic channeling, where intermediates are efficiently passed from one enzyme to the next, reducing losses and side reactions For instance, transferring the withanolide biosynthesis gene cluster from *W. somnifera* into yeast can enable the production of key withanolides by leveraging the yeast’s robust growth and metabolic engineering capabilities. Moreover, co-expression of TFs and regulatory proteins encoded within the gene clusters can optimize the pathway by upregulating or downregulating specific steps as needed [69]. This strategy not only enhances withanolide production but also provides insights into the regulatory mechanisms and interactions within the biosynthesis pathway, facilitating advanced metabolic engineering strategies. These strategies include pathway optimization to maximize flux through the withanolide pathway, enhancing precursor supply by engineering upstream pathways to increase the availability of key precursors like squalene, and minimizing byproduct formation by balancing metabolic flux [53]. Understanding these interactions allows for the development of more efficient and high-yield withanolide production systems in microbial hosts, paving the way for their application in pharmaceuticals, nutraceuticals, and other industries.

## 4. Methods to Enhance the Production of Withanolides by Machine Learning (ML)

### 4.1. Application of ML in Metabolic Pathway and Enzyme Identification

Machine learning (ML) offers promising avenues to enhance the production of secondary metabolites by leveraging its capabilities in data analysis, prediction, and optimization (Figure 2). ML algorithms can analyze large-scale omics data, including genomics, transcriptomics, proteomics, and metabolomics, to identify key genes, enzymes, and regulatory elements involved in secondary metabolite biosynthesis pathways [70]. This deepens our understanding of metabolic networks and helps pinpoint targets for genetic manipulation or pathway engineering to enhance production yields [71]. ML methods have been shown to cause an increase in the production of secondary metabolites in medicinal plants [72]. ML algorithms that consider variables like oxygen levels, pH, temperature, and nutrient availability can forecast the ideal circumstances for microbial or plant cell cultures [73,74]. This predictive power reduces trial-and-error in experimental settings and helps develop more efficient bioprocesses. Additionally, by repeatedly optimizing culture medium compositions and fermentation settings, ML algorithms can maximize the production efficiency of secondary metabolites while reducing resource consumption and waste.

ML methods can be divided into supervised and unsupervised learning approaches that are essential for improving the synthesis of secondary metabolites in plants. Gene expression levels are an example of an input variable used in supervised learning, which forecasts qualitative or quantitative outputs like the production of particular secondary metabolites [70]. To model and anticipate these correlations, a variety of algorithms are used, including neural networks, decision trees, random forests, support vector machines, and Bayesian networks [75]. This enables the optimization of culture conditions or targeted genetic alterations. Unsupervised learning, on the other hand, looks for underlying patterns and structures in datasets without pre-set results by evaluating untagged data. Principal component analysis, k-means clustering, hierarchical clustering, and other algorithms are used to find patterns or clusters in data, which can help decipher intricate metabolic connections or open up new avenues for the creation of secondary metabolites [76]. The fields of plant metabolic engineering and bioprocess optimization for improved secondary metabolite synthesis have greatly benefited from the application of both supervised and unsupervised learning techniques.

Important processes in metabolic engineering, such as identifying metabolic pathways and essential enzymes, are being revolutionized by ML. It used to be necessary to mine the whole genome and characterize each stage of the metabolic pathway in order to identify the genes and gene clusters involved [77]. This method was time-consuming and labor-intensive. In order to identify novel metabolites and the biosynthetic genes that go along with them, this traditional method frequently entailed the analysis of transcriptomic and genomic data. These genes were then inserted into non-native host cells to facilitate simpler engineering and optimization [78]. However, the uncertainties and problems associated with these traditional statistical methods, such as the insufficient definition of biosynthetic genes and redundancy in pathway functions, make it challenging to create accurate predictive metabolic models. In order to enhance metabolic engineering, ML presents a viable substitute by merging diverse and unorganized multi-omics information, including transcriptomics, epigenomics, microRNAomics, and genomics [79]. For example, a predictive ML model was developed that identified junction genes shared by both metabolic processes and distinguished genes involved in primary and secondary metabolism using gene expression, gene network-based features, epigenetics, gene duplication, and protein domains [80]. DeepRF, a supervised ML model that predicts metabolic pathways from genomic sequences by fusing random forests with deep neural networks, was developed [81]. The difficulties of manually creating metabolic pathways are efficiently addressed by DeepRF, allowing for the study of huge genome sequencing data. DeepRF demonstrated strong performance metrics with accuracy over 97%, recall over 95%, and precision over 99% when evaluated on more than 318,016 instances. In contrast, DeepRF fared better than previous techniques, proving its dependability and potency in the prediction of both known and novel metabolic pathways [81]. Similarly, comprehensive proteomics and metabolomics data combined with ML outperform conventional kinetic models in the accurate prediction of route dynamics in bioengineered systems [82]. By automating the prediction process, this method lessens the need for domain expertise and provides bioengineering efforts with both quantitative and qualitative information. In contrast to conventional models, this approach selects the best predictions by methodically utilizing fresh data to improve them without making assumptions about particular relationships [82]. The findings demonstrate how machine learning may be used to increase biological system engineering’s precision and effectiveness. While ML techniques have not been directly applied to withanolide synthesis in *Withania* species, there is great potential to combine ML with multi-omics datasets such as transcriptomics, microRNAomics, genomes, and epigenomics. ML can provide greater insights into the intricate biosynthetic processes of withanolides by utilizing these extensive datasets. Through this integration, important genes, enzymes, and regulatory components involved in the synthesis of withanolides might be identified, opening the door to focused genetic changes and optimization techniques. As a result, using ML to facilitate the manufacture of withanolides in heterologous systems and in vitro culture platforms may improve both yield and efficiency. These developments would simplify the metabolic engineering procedure, lessen the need for labor-intensive, conventional techniques, and open the door to the scalable and long-term synthesis of these important secondary metabolites.

### 4.2. Application of ML to Improve In Vitro Culture Efficiency

The numerous variables that impact tissue culture make it extremely difficult and time-consuming to improve in vitro procedures with the current methodologies. For example, in vitro secondary metabolite production is affected by several factors, including culture conditions, media composition, and explant type, and determining the right levels of these components is crucial to the secondary metabolite’s synthesis. Thus, there is a lot of opportunity to use novel computational approaches for data analysis in order to create more effective procedures. ML models can simulate different scenarios, providing insights into how changes in culture conditions might affect growth and metabolite production. This predictive capability enables researchers to fine-tune culture parameters more precisely, leading to improved efficiency and consistency. For example, in order to maximize in vitro *Cannabis* growth, different light qualities and sucrose concentrations were optimized [83]. GRNN was the most successful ML model used to predict growth parameters, along with MLP and ANFIS. The best light and sucrose levels were then determined by integrating GRNN with optimization algorithms like GA, BBO, ISA, and SOS, which were then verified in further tests. The accuracy of the projected ideal circumstances was validated by the results, demonstrating ML’s power to improve in vitro culture results and direct focused growth tactics in cannabis tissue culture [83]. In another study, a combination of ML (GRNN and RF) and an optimization algorithm was used for indirect shoot regeneration in *Passiflora caerulea* as a prerequisite protocol for the development of *Agrobacterium* genetic transformation [84]. The integration of ML like GRNN and RF with genetic algorithms (GAs) presents a forward-looking approach to optimizing and predicting in vitro culture systems in *Passiflora* tissue culture. This combined method offers potential solutions to current challenges encountered in tissue culture protocols for *Passiflora* species.

As of now, specific reports applying machine learning approaches to *Withania* plant tissue culture appear limited in the literature. Research on *Withania* often focuses on phytochemical analysis, medicinal properties, and agronomic practices rather than integrating machine learning directly into tissue culture optimization. However, given the advancements in ML applications across plant biotechnology, future studies may explore ML’s potential to enhance *Withania* tissue culture protocols for improved production of bioactive compounds.

### 4.3. Use of CRISPR-Based Genome Editing

The CRISPR/Cas genome editing technique has shown promise as a transformative tool for improving the synthesis of secondary metabolites in therapeutic plants. Alkaloids, flavonoids, terpenoids, and phenolics are examples of metabolites, which are important substances with a wide range of medicinal uses. In order to improve these metabolites, traditional breeding techniques are frequently less accurate and time-consuming. A focused, effective, and adaptable method for changing the genes involved in pathways leading to the manufacture of secondary metabolites is provided by CRISPR/Cas [85]. However, only a few therapeutic plants have been altered since there are inadequate transformation and regeneration protocols, whole genome and mRNA sequences, and CRISPR/Cas genome editing tools. Before CRISPR/Cas technology was developed, several genetic editing strategies were employed to achieve considerable advances in the production of medicinally useful specialized metabolites, including gene silencing, gene stacking, and overexpression. These techniques allowed for the enhancement of specific metabolic pathways, which increased the production of desired compounds in medicinal plants, but they lacked the precision and efficiency that CRISPR/Cas technology now offers in genome editing and metabolic engineering [86]. Many different targeted genetic alterations may be induced by it, such as fragment deletion, targeted gene insertion, frameshift and premature termination codon, gene replacement, and site-specific single-gene knockout. The purpose of CRISPR/Cas is to precisely modify the genome in order to find out how individual genes, gene clusters, and their regulatory components work in the biological processes that affect plants. For example, multiplexing, or the simultaneous editing of many genes, is made possible by CRISPR/Cas and is especially useful for improving whole metabolic pathways. This strategy may cause intermediary metabolites to accumulate and reroute flow in the direction of the intended final product. As shown in *A. thaliana* and *N. tabacum* cell cultures, it allows for the simultaneous targeting of several genes, which facilitates the identification of computationally predicted genes implicated in metabolic processes [87,88]. CRISPR techniques make it simple to multiplex and change gRNA coding sequences, which enables effective combinatorial and systematic editing [89]. The cost of screening and library creation is decreased using pooled gRNA techniques. Furthermore, by generating DNA double-strand breaks, CRISPR/Cas-mediated chromosomal dissection may produce targeted chromosomal inversions, deletions, and rearrangements that help with gene cluster identification and metabolic circuit alteration. *N. benthamiana* and *A. thaliana* have been subjected to this technique for advanced genetic research [90].

#### 4.3.1. CRISPR/Cas Multiplex Gene Editing

Plant biotechnology is undergoing a revolution thanks to CRISPR/Cas multiplex gene editing, which provides accurate and effective methods for modifying the genomes of a variety of species, including therapeutic plants. With the use of this technique, it is possible to target numerous genes at once, which is very useful for plants whose complex features are controlled by polygenic interactions. Researchers may introduce specific mutations, deletions, insertions, or substitutions in numerous loci during a single genetic modification event by utilizing the adaptability of CRISPR/Cas systems (Figure 3A). This strategy has enormous promise for strengthening resistance to diseases and environmental challenges, increasing the synthesis of important bioactive chemicals in medicinal plants, and speeding up breeding initiatives. Multiple genes can be edited simultaneously to fine-tune metabolic pathways, increasing the production of medicinal chemicals and facilitating the discovery of new compounds. For example, *Salvia miltiorrhiza*, a plant known for its medicinal properties, specifically its lipid-soluble compounds like tanshinones and water-soluble phenolics like rosmarinic acid (RA), salvianolic acid B (SAB), and salvianic acid, uses CRISPR/Cas9 to perform multiplex gene knockout targeting the laccase gene family [91]. The multicopper-oxidase gene family, which includes laccases, is crucial for the oxidation and polymerization of monolignols, which is necessary for the production of SAB. Twenty-nine laccases (SmLACs) were found in the *S. miltiorrhiza* database by the study. Single-gene knockouts would not provide enough insight into the roles of these genes because of the functional redundancy within the gene family. As a result, a “dual-locus editing” technique to concurrently target many laccase genes was used. Subsequent investigation into the metabolism of phenolic acid revealed a significant decrease in the expression of important genes involved in the production of phenolic acid. This decrease brought attention to the role that *Sm*LAC genes play in the biosynthesis of phenolic acids, which is crucial for root growth and phenolic acid metabolism [91]. Overall, this work showed how effective CRISPR/Cas9 multiplex editing is in identifying the precise functions of highly redundant gene families, offering important new understandings of the metabolic pathways in *S. miltiorrhiza*. A useful method for examining and modifying the activities of highly redundant gene families in plants is CRISPR/Cas9 multiplex gene editing. The researchers were able to overcome functional redundancy constraints by editing numerous genes at once, which improved their comprehension of the functions of the various genes within the family. Despite the success, similar research has not yet been conducted on *Withania* plants, and several factors could contribute to this gap. The above study provides a blueprint for applying similar techniques to *Withania* plants. It may be possible for researchers to reduce functional redundancy and increase the synthesis of these beneficial chemicals by focusing on several genes involved in withanolide biosynthesis. This method’s insights can help optimize metabolic pathways, increasing the yield of therapeutic chemicals. This approach may also make it easier to find new regulatory mechanisms in *Withania*. All in all, it paves the way for sophisticated genetic engineering targeted at enhancing the therapeutic potential of *Withania* plants.

#### 4.3.2. Overexpression and Targeted Gene Knockout

One approach that shows promise for increasing withanolide production, especially in in vitro cultures, is the overexpression of genes encoding enzymes involved in withanolide biosynthesis. Apart from concentrating on enzymes, the production of withanolides can be considerably increased by overexpressing and/or activating genes that encode particular TFs. According to earlier studies, *W. somnifera* exhibits differential expression of TFs such as WRKY and MYB [61,62,92], which are essential to the withanolide biosynthetic pathway. Therefore, these TFs make great targets for genetic modification with the goal of boosting withanolide output. CRISPR-based methods provide an effective means of achieving this objective. Researchers can accurately modify the genome to overexpress or activate these important TFs and enzymes involved in the biosynthesis of withanolides by using CRISPR/Cas systems.

Another approach could be the editing of genes involved in withanolide biosynthesis that are epigenetically controlled (Figure 3B). By regulating gene expression without changing the DNA sequence, epigenetic regulation is essential for managing the synthesis of secondary metabolites in plants. The production levels of secondary metabolites can be impacted by mechanisms that activate or repress genes involved in secondary metabolite biosynthesis, including DNA methylation, histone modification, and RNA interference [93]. Variability in metabolite synthesis can be caused by these epigenetic modifications, which can be impacted by developmental phases, environmental factors, and stressful situations. A way to accurately modify these epigenetic marks is provided by CRISPR/Cas technology. Researchers can target specific gene loci to add or remove methyl groups or modify histones, thereby activating or silencing genes involved in secondary metabolite pathways, by fusing CRISPR/Cas systems with epigenetic modifiers.

CRISPR/Cas9-based gene disruption methods have been widely applied in medicinal plants to modulate the concentration of specific metabolites. This approach can either enhance the production of pharmaceutically beneficial compounds or reduce the levels of harmful metabolites. Through targeted mutagenesis, researchers can precisely edit genes involved in key metabolic pathways, leading to improved medicinal properties and the safety of plant-derived compounds. For example, in order to increase the concentration of helpful compounds and decrease the concentration of poisonous ones in medicinal plants, CRISPR/Cas9-based gene disruption approaches have proven to be useful. For example, altering the hyoscyamine 6β-hydroxylase (AbH6H) gene in *Atropa belladonna* resulted in plants devoid of scopolamine and anisodamine, therefore elevating the level of hyoscyamine, a substance useful in the treatment of organophosphate poisoning and arrhythmias [94]. Targeted mutagenesis of lignocellulose biosynthesis-related enzymes in *Dendrobium officinale* has similarly shown great editing efficiency, improving the pharmacological characteristics of the plant. This shows how CRISPR/Cas9 may be used to improve metabolite profiles and therapeutic plant qualities.

CRISPR/Cas-based gene editing holds immense potential for enhancing the production of withanolides and other secondary metabolites in medicinal plants. However, similar research has not yet been conducted on *Withania* plants. The lack of genomic resources, such as a well-annotated genome and functional gene databases, presents a significant challenge. Moreover, the optimization of CRISPR/Cas protocols for *Withania*, including tissue culture and transformation techniques, may require substantial time and investment. The complexity of withanolide biosynthesis pathways and potential regulatory mechanisms unique to *Withania* might also pose additional hurdles. However, the future scope for *Withania* in this regard is promising. Advancements in genomic and biotechnological tools, including the development of comprehensive gene libraries and efficient transformation protocols, will pave the way for successful CRISPR/Cas applications.

## 5. Conclusions

Gaining a deeper understanding of the withanolide biosynthesis pathway is essential to optimizing withanolide production through metabolic engineering, in vitro cultures, and heterologous systems. When combined with multi-omics data, including transcriptomics, proteomics, metabolomics, and genomes, ML can provide deep insights into the complex networks that control these metabolic pathways. Large datasets can be analyzed by ML algorithms to find important enzymes, metabolic intermediates, and regulatory genes involved in the generation of withanolide. Researchers can identify specific targets for precise manipulation by sorting through these intricate networks. Once modified, the targets can be edited utilizing genome editing techniques based on CRISPR. Many of the biosynthetic genes involved in the formation of withanolides in callus culture are silent because of the cells’ undifferentiated state, which results in lower metabolic complexity. As a result, the biosynthetic pathway can be systematically altered and/or regulated in cell suspension culture through the application of CRISPR-based techniques. In order to enhance withanolide production, specific conditions, such as the type and concentration of elicitors, nutrient composition, and physical parameters like light and temperature, must be optimized. Another promising system for withanolide synthesis is hairy root culture, where withanolide production can be enhanced through CRISPR-mediated genome editing in hairy roots. Standard procedures, including the use of specific elicitors and the optimization of growth conditions, can systematically regulate the generation of withanolides in both hairy root culture and cell suspension culture.

But there are a few issues that must be resolved. Improving production requires locating and removing metabolic bottlenecks in the withanolide biosynthesis pathway. Unintended consequences can also result from inadequate alterations and off-target effects during CRISPR-Cas9 genome editing. Improved CRISPR variants with greater specificity and efficiency are one example of how genomic tool development will continue to improve genome editing precision and enable more successful metabolic engineering.

Prospective futures comprise the amalgamation of machine learning (ML) and artificial intelligence (AI) with omics data to furnish more profound discernments into biosynthetic pathways and regulatory networks, hence permitting more focused and efficacious metabolic engineering tactics. Through the use of these cutting-edge instruments, scientists may create scalable and sustainable withanolide production processes. Withanolide yields could also be greatly increased by investigating various microbial hosts and their genetic modifications for improved precursor supply, such as mevalonate pathway optimization or the introduction of feedback-resistant enzymes. It will be essential to look into metabolic engineering techniques to balance the flux toward withanolide production while reducing byproduct generation. All things considered, the development of synthetic biology, data integration, and genomic techniques holds enormous potential for overcoming these obstacles and producing withanolides in a way that is both scalable and sustainable. By tackling these obstacles and utilizing novel technologies, the domain can progress towards more effective and financially feasible techniques for producing these precious substances.

## 6. Future Prospects

In order to advance withanolide production in the future, a number of important research and development areas should be given top priority. First, it is crucial to keep researching CRISPR-based methods for optimizing the biosynthetic pathways in microbial and plant systems. In order to activate or silence particular genes, this involves not just gene editing but also the targeted epigenetic alterations achieved using CRISPR. Second, to maximize withanolide yields in cell suspension and hairy root cultures, adjust culture conditions, such as nutritional medium. By combining AI and ML to examine omics data, it will be easier to find new metabolic bottlenecks and regulatory elements, opening up new avenues for genetic and metabolic engineering research. Furthermore, the translation of laboratory results into commercial-scale production methods will require close cooperation between academic institutions, industry, and regulatory agencies. 

In order to address the increasing demand for withanolides in the pharmaceutical, nutraceutical, and other industries, it will be important to develop robust and scalable bioprocessing procedures, including downstream processing and bioreactor design. Through tackling these domains, researchers can guarantee the prosperous and enduring manufacturing of withanolides, ultimately culminating in progress within the healthcare and allied fields.

## Figures and Tables

**Figure 1 plants-13-02171-f001:**
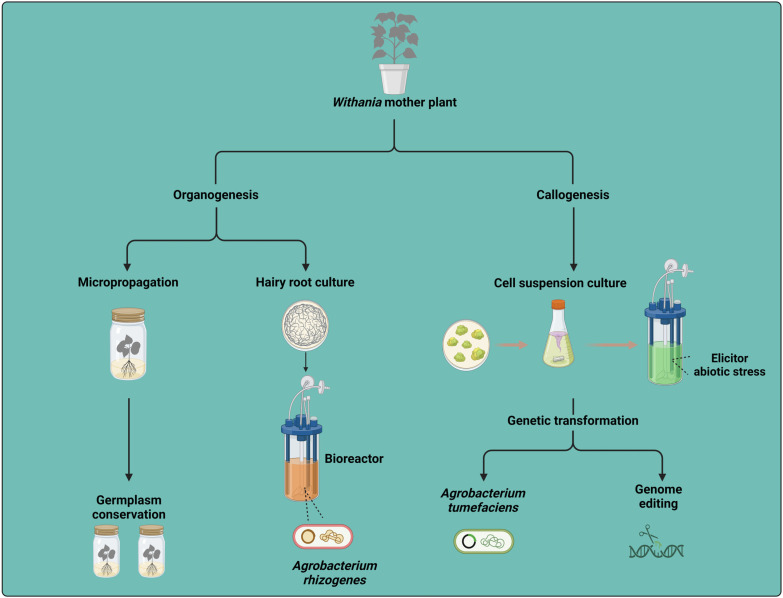
The conceptual framework for withanolide synthesis and modification involves utilizing in vitro culture techniques and heterologous expression systems. This approach allows for the production and enhancement of withanolides through controlled manipulation of plant cells or tissues in a laboratory setting, as well as by expressing withanolide biosynthetic genes in alternative host organisms. The figure was created with BioRender.com.

**Figure 2 plants-13-02171-f002:**
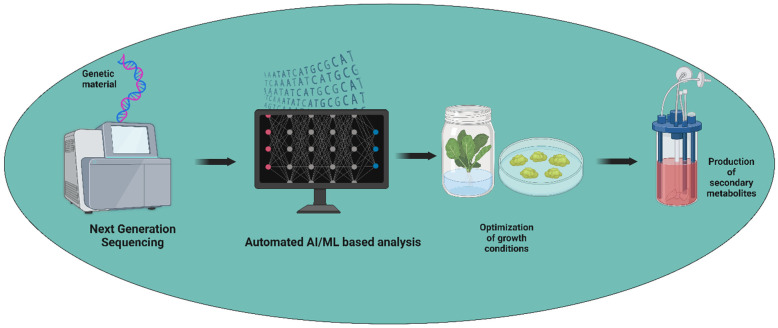
The application of automated artificial intelligence/machine learning for withanolide production. The figure was created with BioRender.com.

**Figure 3 plants-13-02171-f003:**
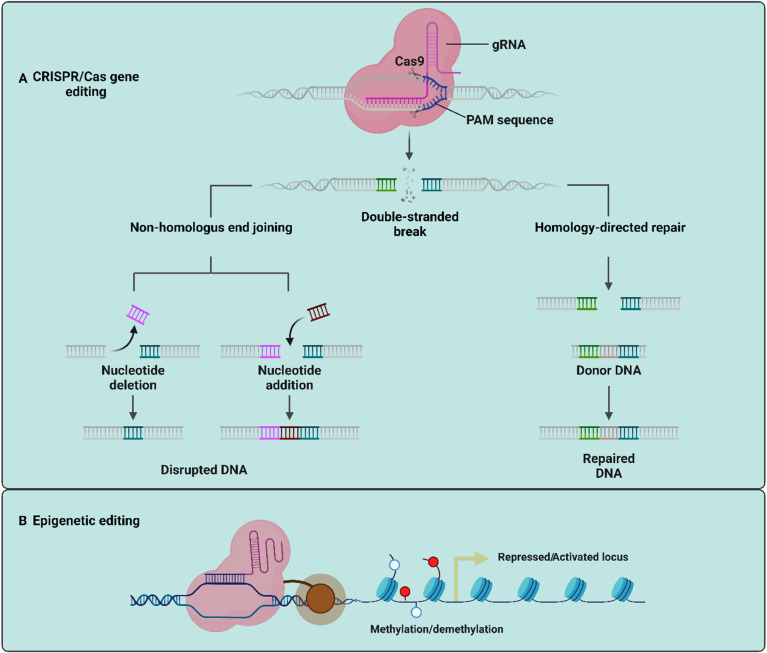
CRISPR-based genome and epigenome editing. (**A**) The Cas endonuclease is used to cleave specific sites on double-stranded DNA using single-guide RNAs (sgRNAs) that bind to target sequences. The presence of a protospacer adjacent motif (PAM), a short region in the genome where CRISPR RNA binds, is necessary for Cas9 to recognize the target sequence. When a double-stranded break (DSB) is created, one of two plant DNA repair pathways is activated. Homology-directed repair (HDR) leads to point mutations or gene replacement by using a similar DNA donor template. NHEJ, on the other hand, is more error-prone and tends to produce small insertions or deletions. (**B**) This figure illustrates the application of CRISPR technology in epigenetic regulation through targeted methylation and demethylation. The CRISPR-Cas9 system is depicted as guiding epigenetic modifiers to specific gene loci. This targeted approach allows for precise modulation of gene expression by altering the epigenetic state of specific genomic regions.

**Table 1 plants-13-02171-t001:** Withanolide production through in vitro culture.

Explant	Culture Type	Type of Elicitor	Metabolite Content	Reference
Leaf	Callus culture	Chitosan at 100 mg/L	27-OH-withanolide-A (52.0 µg/mL, 160%)	[33]
Leaf	Callus culture	UV rays for 60 and 90 min	Withanolide-A (241.1 µg/mL, 99% increase) and 27-OH-withanolide-B (124.8 µg/mL, 40% increase)	[33]
Root	Cell suspension culture	Mycelial extract (1% *w*/*v*) and culture filtrate (5% *v*/*v*) of the endophytic fungus *Aspergillus terreus* 2aWF	Withanolide A (12.20 ± 0.52 µg/g FCB) at 6 h. Withanolide A (10.29 µg/g FCB withanolide A) at 24 h.	[34]
Shoot	In vitro shoot culture	MS basal medium supplemented with BAP (4.44 µM), 3% sucrose, and 0.8% agar	Withanolide A (129.18 ± 0.33) and withaferin A (968.6 ± 0.45)	[34]
Leaf	Cell suspension culture	Organic additive, seaweed extracts (Gracilaria edulis)	Withanolide A (7.21 mg/g), Withaferin A (4.23 mg/g)	[16]
Leaf	Adventitious root cultures	Salicylic acid at 150 μM	64.65 mg g^−l^ DW withanolide A (48-fold), 33.74 mg g^−l^ DW withanolide B (29-fold), 17.47 mg g^−l^ DW withaferin A (20-fold), 42.88 mg g^−l^ DW withanone (37-fold),	[35]
Leaf	Hairy root cultures	4% sucrose	Withanolide A production (13.28 mg/g DW)	[36]
Nodal explant	In vitro shoot culture	Polyamine at 20 mg/L	6.5 times (leaf), 3.3 times (root)	[37]

**Table 2 plants-13-02171-t002:** The use of synthetic biology for withanolide production.

Technique Utilized	Gene/Enzyme Involved	Purpose of Modification	Experimental Details	Outcomes/Benefits	References
Heterologous	*CYP749B1*; *CYP76*; *CYP71B10*	Enhance withanolide production	Enhanced expression upon MeJ treatment and VIGS-based overexpression/silencing were conducted in *W. somnifera* and transgenic tobacco.	*WsHMGR2* overexpression increased cycloartenol, sitosterol, stigmasterol, and campesterol, while *WsDXR2* transgenic lines reduced cholesterol, indicating the MVA pathway’s preference for withanolide production.	[57]
Heterologous	*CYP71B35*	Enhance withanolide production	WsCYP71B35, induced by MeJ, catalyzes withaferin A, withanolide A, withanolide B, and withanoside IV in yeast microsomes.	Overexpression of *WsCYP71B35* in *W. somnifera* and transgenic tobacco increased withanolide levels and resistance to *P. syringae*, confirming its role in metabolite production and plant defense.	[58]
Microbial heterologous (*E. coli*)	Squalene Synthase (WsSQS)	Enhance withanolide biosynthesis	Recombinant WsSQS produced in *E. coli*; highest expression in leaves; promoter analysis revealed cis-acting elements; upregulation by SA, MeJ, and 2,4-D	High expression in leaves; insights into regulatory elements and spatial/temporal regulation of withanolide biosynthesis	[59]
Microbial heterologous (Yeast)	ERG4 and ERG5 enzymes replaced with DHCR7	Production of 24-methylene-cholesterol, a precursor for physalin and withanolide	Flask-shake culture; evaluated three DHCR7 versions from various organisms; XlDHCR7 showed the best efficiency	Efficient methylene-cholesterol production; viable platform for further research into downstream enzymes	[60]

## Data Availability

No new data were created.

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
