# Peer review of "Enhancing Withanolide Production in the Withania Species: Advances in In Vitro Culture and Synthetic Biology Approaches"

_plants, 2024, doi:10.3390/plants13152171_

Round 1

Reviewer 1 Report

Comments and Suggestions for Authors

Dear Authors,

congratulations for the work. The review is quite interesting. I work with isolationg of withanolides and it was very informative to read this manuscript. However, in oder to increase the interest of the readers I strongly suggest the authors to insert some tables resuming the information described n the text.

- It would show rapidly the most important data found by the authors in the literature sources. So I suggest some tables in the sections to resume the quantitative data, showing to the readers how the Biotic or abiotic elicitors, feeding precursors increase the metabolism, with strategies, problems, ad yielding processs, or adevantages and desadvantages between the options suggested. It would be very interest to see how the extrenal stimuli really contribute quantitatively to the eancrase withanolides production in numbers. Not only say the the synthesis were augmented or not. 

Ex: I would like to know the concentrations used in this example:

"This stress stimulates pathways involved in secondary metabolite biosynthesis as part of the plant's defense mechanism.  For example; a range of NaCl concentrations significantly increased withaferin A, withanolide A and withanone levels in shoots and roots compared to the control in W. somnifera [27]. By upregulating important biosynthetic genes under NaCl stress, melatonin promoted the production of withanolides, which resulted in an increase in the accumulation of withanolides A and withaferin A in W. coagulans [28]."

- I also suggest they could insert some pharmacological figures correlating the enzimes up adn down regulation, and the biosynthesis effect of this process. Which enzime are the key in the withanolide biosynthesis and what compounds could inhibit or stimulate its products, create a figure about it. It would let the text less tedious and atract the readers attention.

- Pls, pay attention with the long text with just one reference being cited, and also the following sentenses with the same citation one imediately after another, which is not good. Ex:

"Precursor feeding in cell suspension cultures for enhanced 142 production of secondary metabolites like withanolides is often limited due to several factors. Optimizing the concentrations and timing of precursor addition requires detailed understanding of the metabolic pathways involved, which is not well-characterized for genus Withania. In addition, identifying suitable precursors that can effectively stimulate the biosynthetic pathways without causing cytotoxic effects or metabolic imbalances can be challenging. In a study, optimization of major and minor withanolides in W. somnifera 148 was carried out using elicitors and precursor feeding strategies [19]. The highest withanolide yields were achieved with chitosan and squalene, along with picloram, Kn, L-glutamine, and sucrose, in bioreactors. This protocol resulted in significantly higher withanolide concentrations compared to controls, both in shake-flask and bioreactor cultures [19]. Precursor feeding presents difficulties because of metabolic complexity, expense, optimization, and possible toxicity; nevertheless, these obstacles may be overcome with further study and technical advancements. Subsequent developments in metabolic engineering, biotechnological instruments, and industrial use are anticipated to broaden the range and significance of precursor feeding in augmenting the yield of important chemicals generated from plants."

and

"In a recent study, the effects of SA and cellulase from Aspergillus niger was used  in W. coagulans cell suspension cultures, wherein, SA was found more effective and promote higher high withanolides accumulation than cellulase [21]. The results revealed that all the both the elicitors at all durations of treatment promoted total phenol and flavonoid content, Withaferin A and Withanolide A accumulation in leaf and stem derived cultures. However, the higher levels of accumulation for Withaferin A and Withanolide A was observed in the cell suspension cultures derived from the leaf explant in comparison to the stem derived cultures [21]

- Pls, also rewrite the references:

most of them are writen showing just the namber of the first author + et al.

"Parvin, S., et al., Potential Role and International Trade of Medicinal and Aromatic Plants in the World. 840 European Journal of Agriculture and Food Sciences, 2023. 5(5): p. 89-99.

Author Response

Reply for reviewer 1; [Highlighted in yellow]

Comment#1

congratulations for the work. The review is quite interesting. I work with isolationg of withanolides and it was very informative to read this manuscript. However, in oder to increase the interest of the readers I strongly suggest the authors to insert some tables resuming the information described n the text.-

It would show rapidly the most important data found by the authors in the literature sources. So I suggest some tables in the sections to resume the quantitative data, showing to the readers how the Biotic or abiotic elicitors, feeding precursors increase the metabolism, with strategies, problems, ad yielding processs, or adevantages and desadvantages between the options suggested. It would be very interest to see how the extrenal stimuli really contribute quantitatively to the eancrase withanolides production in numbers. Not only say the the synthesis were augmented or not.

Answer

We agree with this comment. We added a table dealing with explant type, in vitro strategies, type of elicitors used for the enhancement of the withanolides.

Comment#2

Ex: I would like to know the concentrations used in this example: "This stress stimulates pathways involved in secondary metabolite biosynthesis as part of the plant's defense mechanism. For example; a range of NaCl concentrations significantly increased withaferin A, withanolide A and withanone levels in shoots and roots compared to the control in W. somnifera [27]. By upregulating important biosynthetic genes under NaCl stress, melatonin promoted the production of withanolides, which resulted in an increase in the accumulation of withanolides A and withaferin A in W. coagulans [28]."

Answer

Thank you for pointing out. We added the quantitative values and details with the findings.

Comment#3

- I also suggest they could insert some pharmacological figures correlating the enzimes up adn down regulation, and the biosynthesis effect of this process. Which enzime are the key in the withanolide biosynthesis and what compounds could inhibit or stimulate its products, create a figure about it. It would let the text less tedious and attract the reader’s attention

Answer

Thank you for your valuable suggestion. While we currently do not have the figures correlating enzyme regulation and the biosynthesis effects of this process, we agree that including such figures would greatly enhance the manuscript. We will work on generating these figures to illustrate the key enzymes involved in withanolide biosynthesis and the compounds that could inhibit or stimulate their production.

However, a separate section 3.2.1 Biosynthesis gene clusters for withanolides has been added. Dealing with the key enzymes involved in withanolide production.

Comment#4

- Pls, pay attention with the long text with just one reference being cited, and also the following sentenses with the same citation one imediately after another, which is not good. Ex:

"Precursor feeding in cell suspension cultures for enhanced 142 production of secondary metabolites like withanolides is often limited due to several factors. Optimizing the concentrations and timing of precursor addition requires detailed understanding of the metabolic pathways involved, which is not well-characterized for genus Withania. In addition, identifying suitable precursors that can effectively stimulate the biosynthetic pathways without causing cytotoxic effects or metabolic imbalances can be challenging. In a study, optimization of major and minor withanolides in W. somnifera 148 was carried out using elicitors and precursor feeding strategies [19]. The highest withanolide yields were achieved with chitosan  and squalene, along with picloram, Kn, L-glutamine, and sucrose, in bioreactors. This protocol resulted in significantly higher withanolide concentrations compared to controls, both in shake flask and bioreactor cultures [19]. Precursor feeding presents difficulties because of metabolic complexity, expense, optimization, and possible toxicity; nevertheless, these obstacles may be overcome with further study and technical advancements. Subsequent developments in metabolic engineering, biotechnological instruments, and industrial use are anticipated to broaden the range and significance of precursor feeding in augmenting the yield of important chemicals generated from plants.

and

"In a recent study, the effects of SA and cellulase from Aspergillus niger was used in W. coagulans cell suspension cultures, wherein, SA was found more effective and promote higher high withanolides accumulation than cellulase [21]. The results revealed that all the both the elicitors at all durations of treatment promoted total phenol and flavonoid content, Withaferin A and Withanolide A accumulation in leaf and stem derived cultures. However, the higher levels of accumulation for Withaferin A and Withanolide A was observed in the cell suspension cultures derived from the leaf explant in comparison to the stem derived cultures [21]

Answer

  • As suggested, two reference has been added.
  • The long sentences have been divided for more clear meaning.
  • The repetition of the same citation immediately after another has been removed from the said paragraph one.
  • Our aim was to add the reference at the beginning and end of the discussion of the same finding, especially if it is lengthy, to keep the reader engaged. Therefore, we did not remove the reference from the second paragraph.

Comment#5

- Pls, also rewrite the references: most of them are writen showing just the namber of the first author + et al.

"Parvin, S., et al., Potential Role and International Trade of Medicinal and Aromatic Plants in the World. 840 European Journal of Agriculture and Food Sciences, 2023. 5(5): p. 89-99

Answer

Thank you for pointing out. We revised the references.

Reviewer 2 Report

Comments and Suggestions for Authors

The manuscript in reference describes a compilation as a mini-review of the progress on in-vitro-based production of withanolides. The review is interesting and offers good and current information for the readership. However, some points should be addressed to make some improvements, additions, and clarifications.

1. The manuscript is practically related to producing withanolides from the Withania species. This must be reflected in the title.

2. Line 17: Revise this statement since withanolides do not strictly occur in withania species. The authors must add a clarification about it. Be consistent throughout the manuscript.

3. Lines 21-23: Improve this passage since it is not entirely understandable.

4. Line 47-48: There is a short connection between the market and a justification about the importance of the Withania species missing.

5. To facilitate information integration for readers, one or two tables summarizing the different reported conditions for cell suspension and hairy root cultures should be added, along with a brief discussion of these tables.

6. Similar to the previous comment, a table gathering conditions and scopes for section 3.0 and its subsections must be added.

7. Section 3.2. A section related to biosynthesis gene clusters for withanolides, beyond specific genes, can be added.

8. Line 765: The caption of this Figure must include more details since it should be self-explanatory. Revise the other Figures to fulfill this quality requirement.

9. Section 5 must be improved since it is challenging to follow. In addition, they are very general and laconically organized. I suggest the concluding remarks must be oriented to define conditions, considerations, and conceptual and experimental implications on cell suspension and hairy roots. Similar to the other sections. Finally, the outlook can be added to a further, previous, separate section.

Comments on the Quality of English Language

The manuscript requires deep, detailed scrutiny to revise various grammar, stylistic, and connectivity issues throughout the manuscript.

Author Response

Reply for reviewer 2; [Highlighted in green]

Comment#1 The manuscript is practically related to producing withanolides from the Withania species. This must be reflected in the title.

Answer

We agree with this comment. Therefore, we have changed the title. Following is the new title.

“Enhancing Withanolide Production in Withania Species: Advances in In Vitro Culture and Synthetic Biology Approaches”

Comment#2 Line 17: Revise this statement since withanolides do not strictly occur in withania species. The authors must add a clarification about it. Be consistent throughout the manuscript

Answer

We agree with this comment. Therefore, we have re-write the sentence for more clear meaning. We also mentioned the species as per the findings throughout the MS.

Comment#3 Lines 21-23: Improve this passage since it is not entirely understandable.

Answer

We agree with this comment and therefore we re-write the sentence for more clear meaning.

Comment#4 Line 47-48: There is a short connection between the market and a justification about the importance of the Withania species missing.

Answer

We agree with this comment and therefore we added one paragraph in the same section. Moreover; the information regarding the market value has been already discussed.

Comment#5 To facilitate information integration for readers, one or two tables summarizing the different reported conditions for cell suspension and hairy root cultures should be added, along with a brief discussion of these tables.

Answer

We agree with this comment. We added one table dealing with different reports on cell suspension and hairy root culture.

Comment#6 Similar to the previous comment, a table gathering conditions and scopes for section 3.0 and its subsections must be added.

Answer

We agree with this comment. We added the table.

Comment#7 Section 3.2. A section related to biosynthesis gene clusters for withanolides, beyond specific genes, can be added.

Answer

We agree with this comment. A new section of biosynthesis gene cluster for withanolide has been added.

Comment#8

Line 765: The caption of this Figure must include more details since it should be self-explanatory. Revise the other Figures to fulfill this quality requirement.

Answer

Thank you for pointing out. We revise the figure for more clear meaning.

Comment#9

Section 5 must be improved since it is challenging to follow. In addition, they are very general and laconically organized. I suggest the concluding remarks must be oriented to define conditions, considerations, and conceptual and experimental implications on cell suspension and hairy roots. Similar to the other sections. Finally, the outlook can be added to a further, previous, separate section.

Answer

We agree with this comment. We -rewrite the conclusion section. We separated the future prospects section as suggested.

Comment#10

The manuscript requires deep, detailed scrutiny to revise various grammar, stylistic, and connectivity issues throughout the manuscript.

Answer

Thank you for pointing out. We revise the MS for English and grammar and minimize the connectivity issues.

Round 2

Reviewer 1 Report

Comments and Suggestions for Authors

Dear authors,

the manuscript was improved and the informations are clearer and easier to access now. The insert of tables really help the readers to find the informations in fast way. Nice work. 

Reviewer 2 Report

Comments and Suggestions for Authors

The authors addressed my comments adequately, and the manuscript improved in quality and content. Therefore, it can be accepted in its current form.